# Isolation limits spring pollination in a UK fragmented landscape

**Dongbo Li** **¤\*, Christopher F. Clements, Jane Memmott**

School of Biological Sciences, University of Bristol, Bristol, United Kingdom

¤ Current address: School of Biology, University of Leeds, Leeds, United Kingdom
\* Dongbo.li@bristol.ac.uk

## Abstract

Animal-mediated pollination determines the reproductive success of most flowering plants; this process however can be disrupted by environmental degradation, with habitat loss and fragmentation highlighted as a top driver of pollination deficits. Despite being a pervasive stressor worldwide, we still have rather limited empirical evidence on its effects on pollination services, especially for early spring pollination syndromes. We investigate this using a potted plant phytometry experiment in which we placed English Bluebell (*Hyacinthoides non-scripta*)—a species largely pollinated in spring—into a fragmented woodland habitat. We selected 51 woodland patches which varied in both size and distance from each other and placed 153 pots of bluebell plants in the patches for *c.*4 weeks to measure pollination. The woodlands were located in a matrix of grassland, the latter being of low plant species richness and overall in the patches, woodland plants showed a positive species-area relationship. We collected traits on bluebell reproduction, these included the number, size, quality of seeds, the number of seed capsules and the number of flowers that failed to set any seeds. We found that seed traits responded differently to patch area and isolation. Patch isolation negatively affected the number of seeds and capsules, whilst it did not affect the size and quality of seeds. Patch area had no effect on any traits, suggesting that patch area might not necessarily be a factor that affects pollination in this species. The number of flowers that failed to set seed was unaffected by either patch area or isolation. Our study suggests that woodland fragmentation impacts the pollination of understory spring flowering plants. Our results highlight the use of multiple traits of phytometer plants to evaluate pollination and the importance of connectivity in maintaining pollination services in small-fragmented landscapes.

**Data Availability Statement:** Data and software codes are available in figshare data (https://doi.org/10.6084/m9.figshare.26046196) and GitHub

## Introduction

Animal-mediated pollination is an important ecosystem service providing substantial benefits to both humans and wild populations. Thus 75% of crops and 87.5% of flowering plants rely on animal pollination [1, 2], primarily insects, representing an economic value of US$195–387 billion [3]. In addition, pollination is a key driver to the diversification of plants and their

**Funding:** D. L was funded by the China Scholarship Council (grant no. 20186190011).

**Competing interests:** The authors have declared that no competing interests exist.

associated pollinators, with considerable evolutionary and ecological implications for biodiversity [4].

The loss and fragmentation of natural habitats, primarily due to increased human activities, is widespread in terrestrial ecosystems. There is clear evidence of a decline in pollinators [5] and their interactions with plants [6], and habitat loss and fragmentation are considered as one of the top factors driving this pattern [7, 8]. Habitat loss reduces the diversity and abundance of pollinators, with a strong negative effect on pollination [e.g., 9, 10]; whilst the effect of fragmentation is less clear. Habitat fragmentation is a process by which habitats are broken apart into small patches when total areas of habitats remain the same, causing changes in patch isolation and habitats' amount [11]. In general, large habitat patches can provide resources for more pollinators than small patches and potentially contribute to higher pollination if plants are pollen-limited [12, 13]. In addition, as abiotic factors such as temperature, light, and humidity can be altered near habitat edges, variations in microclimate within fragmented landscapes may create different hotspots for pollinator communities [14, 15]. For example, greater light availability at forest edges could promote better nesting resources to bees, leading to higher abundance and diversity than patch interior [15, 16]. Despite previous studies having shown that habitat loss and fragmentation may affect species and biodiversity differently [17, 18], the concurrent nature of these two processes makes their individual contribution challenging to predict.

The theory of island biogeography provides a conceptual backdrop for understanding the effect of habitat loss and fragmentation [19]. It assumes that habitats are "islands" surrounded by an "ocean" of unsuitable matrix, and that the number of species found on an island increases with the area of the island but decreases with distance to the "mainland". Numerous studies have been undertaken to investigate the effect of fragmentation, where patch area and isolation are used as predictors for the occurrence of species [e.g. 20]. For insect pollinators, patch area and isolation are important variables associated with their activity and distribution. For example, a high abundance of pollinators is more likely to be found in large habitat patches [21], and the abundance and richness of pollinators is negatively related with isolation [22]. A decrease in the availability and occurrence of pollinators due to changes in landscape configuration likely reduces the reproductive fitness of plants, especially for plants that are self-incompatible [23]. To disentangle effect of habitat loss and fragmentation on pollinator-plant interactions, we need to understand how these two processes of fragmentation–changes in patch area and the impact of isolation, affect pollinator activities and thereby affect the reproductive success of plants.

Historically, studies have investigated the impact of habitat loss and fragmentation on pollination using a variety of measures collected from patches, including pollinator visitation [24, 25], changes in pollinator assemblages [26, 27], pollen dispersal [26, 28], and seed and fruit production [13, 29, 30]. In most cases, the plants were growing naturally in these patches (but see [22] and [31] for an exception). Although measuring pollinator diversity and visitations within fragmented landscape can provide useful information on pollination and reproductive outputs of pollinated plants, it often requires intensive field sampling and identification of pollinators. Instead, a cost-effective approach to directly estimate pollination services is using potted plant phytometers [32], where bioassay plants (which are self-incompatible) were grown under uniform conditions and subsequently placed in the field site for a certain duration. Hence the seed sets of these bioassay plants serve as a reliable measure for assessing pollination. In that case, phytometer plants can be also selected to target specific pollinator groups based on their phenology and flower morphology [33, 34]. This approach has previously been used to characterize landscape elements [33, 35, 36], and effect of habitat loss and edge effect on pollination services [31].

Spring flowering plants provide pollen and nectar for overwintering pollinators such as bumblebee queens, and the availability of those resources has been shown important for subsequent colony development [37]. Bumblebee queens are one of the earliest pollinators active in spring in northern Europe, foraging and nesting at the time when most of the pollinators have not emerged. Once queens establish new colonies, the first cohort of new bumblebee workers will be produced and take over foraging in late spring [38]. The timing and relative low abundance of these overwintered pollinators mean that visitation can be difficult to observe [39]. Moreover, as habitat metrics and microclimate are associated with the spatial distributions of available nesting resources, it may be critical to the activities of early spring pollinators. For instance, forest edges could provide suitable nesting and mating sites for cuckoo bees (*Nomada sp*.) and mining bees (*Andrena sp*) in spring [40, 41]. Small fragments and forest edges could facilitate more bees and hoverflies in spring, due to rich understory floral resources provided before canopy closure [42]. This stresses the needs to consider the landscape-level impact on spring pollination, as spring flowering plants not only are more susceptible to pollinator limitation [43, 44], but can respond strongly to climate change [45]. However, to our knowledge, there are relatively few fragmentation studies focusing on spring pollination.

Here we investigate the effect of patch area and isolation on pollination in a naturally fragmented landscape, using the English bluebell (*Hyacinthoides non-scripta*), a pollinator-limited plant species largely pollinated by early spring bumblebee queens in the UK [46] as a phytometer plant. We selected 51 woodland patches that varied in size and distance from each other, and placed bluebell plants on the selected patches under fully replicated conditions. We surveyed the plant species found in the woodland patches and measured the seed set and seed capsules of bluebells to quantify the effect of patch area and isolation on pollination.

## Methods and materials

### Study site

Our experiment was conducted between April to May 2022 at Durdham Downs, Bristol, United Kingdom (51.4661 N, -2.6237W), under a field access issued by the nature conservation office at Bristol City Council in February, 2022. The 1.7km$^2$ study site mainly consists of urban grassland, with numerous patches of woodland and scrub scattered within this area (Fig 1 and S1 Fig in S1 File). The grassland is either mown regularly or managed as hay meadows, whilst the woodland and scrub patches are left largely unmanaged. This area is a part of the Special Area of Conservation in Avon Gorge, which is recognized as a Site of Scientific Interest (SSSI) in the UK. During our experiment, very few grassland species were flowering, and the matrix is of very limited value to pollinators. The most common plant species in the woodlands are ash (*Fraxinus excelsior*), oak (*Quercus robur*), hawthorn (*Crataegus monogyna*) and elder (*Sambucus nigra*). Spring-flowering plants found in the woodland and along the woodland edge are wild cherries (*Prunus avium*), cow parsley (*Anthriscus sylvestris*), hogweed (*Heracleum sphondylium*), and alexanders (*Smyrnium olusatrum*). These species can attract a range of spring pollinators including flies, bees, and beetles. Bumblebees such as *Bombus terrestris* are common pollinators in early spring in the southwest UK [37].

### Proximity index

A total of 51 woodland patches that varied in size and shape were selected as experimental fragmented landscape (Fig 1). The edges of patches were marked according to the location of the tree canopy in QGIS [v 3.22, 47], using the most recent version of Google satellite map with an accuracy of 2m [48]. We measured the areas and the edge-to-edge distance between all the patches, and used a proximity index [49] to examine the degree of isolation from a focal

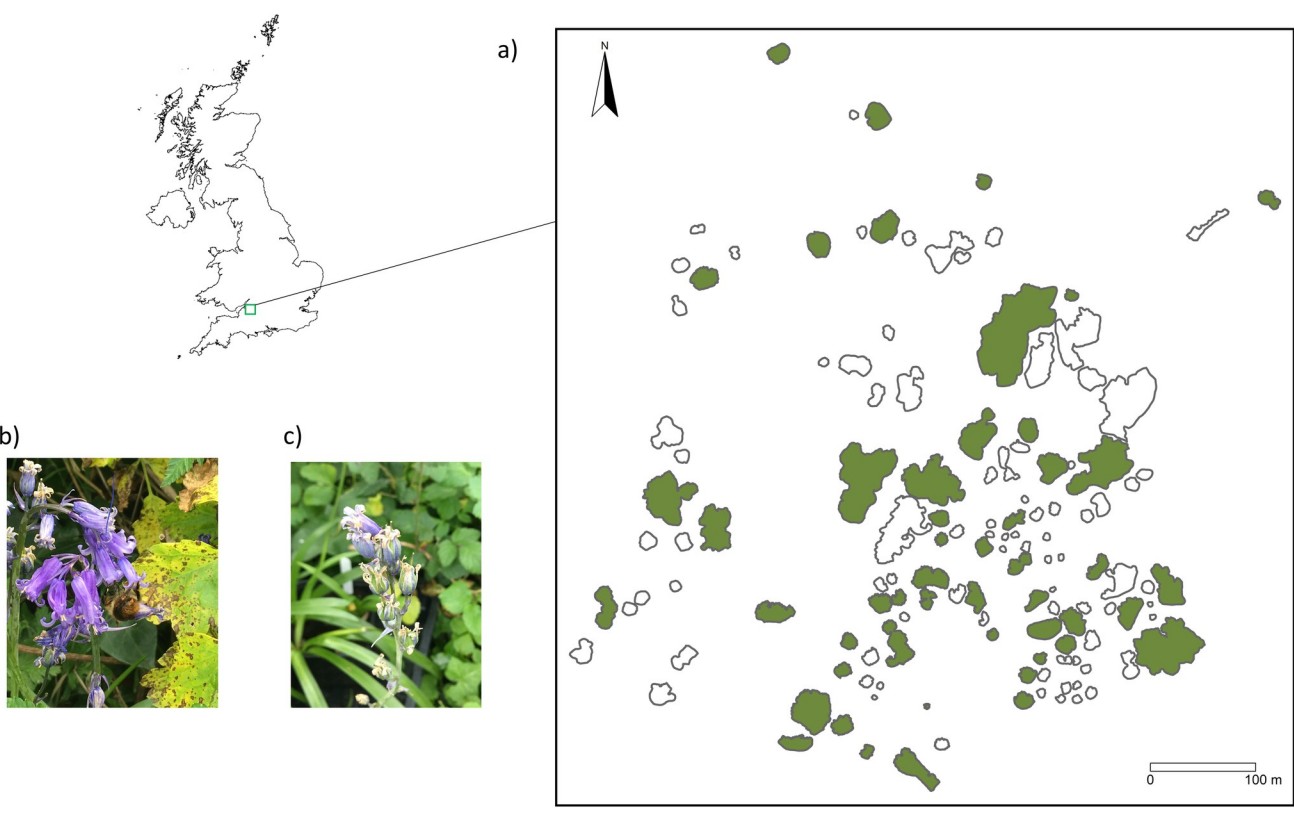

**Fig 1.** (a) The layout of the 51 woodland patches used for the field experiment (n = 51, green) and other surrounding patches (n = 81, unfilled). Map of study site was created by authors using QGIS v3.22 [47]. Basemap was acquired from 'GADM' open-access data (https://gadm.org/). (b) Bluebells were pollinated by bumblebees (*Bombus spp.*). (c) Seed capsules of bluebell, *H. non-scripta*.

patch to the rest of all neighbouring patches. The proximity index is the area-based index weighted by distance [50], so that

$$Px = \sum_{i=1}^{n} A_j * D_{ij}^{-1}$$

For focal patch *i*, A*j* is the area of a neighbour patch *j*, D*ij* is the nearest edge-to-edge distance between a focal patch *i* and a neighbour patch *j*, and n is the total number of patches. The value of the index is larger when a patch is surrounded by a cluster of larger and/or closer neighbour patchers, indicating a lower degree of isolation. We included all the patches in the calculation of Px because bumblebees are expected to forage within a whole landscape [51].

## Plant surveys in woodland patches

We collected data on plant species in the selected woodland patches to investigate species-area relationship more generally in our landscape, including number of woodland species found per patch, and if, any wild variant of English bluebells occurred in the landscape. Field surveys of plants in the selected patches were conducted in late March 2022, a period when bumblebee queens were starting to forage. We conducted a full-patch plant survey by taking random samples of all ground vegetation, including both woody species, herbs and grass, to record the overall plant richness of each patch. To account for the variation in the numbers of spring

flowering plants among woodland patches, we recorded whether each plant species was flowering or not (see *Supp. Info.*). Plants were identified to species level.

## Placement of plants

Bluebell (*H. non-scripta*) is a native plant species, mostly found in the woodland understorey in the UK. Bluebells flower in spring and are largely pollinated by early spring pollinators such as bumblebees [52]. Its flowers show a degree of self-incompatible and pollinator-limited, producing more seeds and seed capsules after cross-pollination [53]. Three bluebell bulbs were grown in a plant pot as experimental units and kept isolated from pollinators before being placed in the field. All the potted plants were kept in the same growing conditions until they started to flower.

Most potted plants started to flower roughly at the same time. We placed plants in each woodland patch at the same time with their flower size randomized (i.e. each batch contained a mixture of flower sizes). Three pots of plants were placed in a tray so they could be easily watered, and trays (each containing 3 pots and a total of 9 plants) were placed in the understory of each woodland patch (regardless of their size and isolation), avoiding patch edges. Placing 3 pots of 9 plants per patch may result in cross pollination between plants in the same patch, but this should provide enough replicates if some of plants were lost or finished flowering too early. Plants were placed outside when they were about to flower, and remained outside until most of plants were had finished flowering, which took approximately four weeks, resulting in a total of 130 pots of *c*. 390 plants collected at the end of the experiment. Two of the total 51 patches had completely not pots of plants collected at the end of experiment, and consequently these two patches were excluded from further analysis. Plants were then returned to an unheated greenhouse to keep them free of pollinators and a muslin bag was placed over each of the flowering racemes when flowering finished, to facilitate the collection of the seeds.

## Collecting seed set data

We measured multiple traits of seeds and seed capsules related to pollination. Specifically, we measured the number of seeds, the quality of seeds (defined as the average weight per seed × 1000), the size of seeds, the number of fruit capsules, and the number of flowers that failed to set any seeds. The number and size of seeds are related to how much pollen was brought to stigma by pollinators, and the number of seed capsules and failed flowers have previously been shown to be positively and negatively related to cross pollination [53]. The quality of seeds was calculated by weighing the dry biomass of seeds collected from one pot of plants, which was then divided by the number of seeds from the same pot. The seed size was measured by randomly taking a representative portion of total seeds collected per pot as a sub-sample, with the number of seeds measured per portion ranged from 1 to 187 (small subsamples were used when total seeds per pot were low). Those samples were photographed using a lab camera (Nikon D5200 18-105mm), and taking sub-samples allowed all seeds to be photographed in a high image quality. The photographs were then processed using Fiji software [54] to calculate the average seed diameter within each sample. To quantify the effect of open pollination, six pots with three bluebell plants in each, were kept separate from pollinators in greenhouse, providing a comparison to open pollination.

## Data analysis

To evaluate the species-area relationship, we used a linear model to fit data on the total number of plant species, and total number of spring flowering species surveyed in the woodland patch, incorporating the logarithm of patch area as a predictor. The slopes (z) of linear models were

used to compare the steepness of the relationship between species and area, and adjusted R-squares were used to evaluate the goodness of model fit.

To investigate the joint effects of patch area and isolation on pollination rates, we used data on (1) the number of seeds, (2) the quality of seeds (i.e., measured as averaged weight per seed), (3) the size of seeds, (4) the number of seed capsules, and (5) the number of undeveloped flowers from each pot. Because of overdispersion, we averaged the data on the number, size, quality of seeds, number of seed capsules, and number of undeveloped flowers collected from the three bulbs grown in each experimental unit to provide a mean measure per woodland. As there were different numbers of spring flowering plants found in woodland patches, which may affect pollinator activities, we standardized patch area by the number of spring flowering species. We then fitted a GLM with a Gaussian distribution on the mean value of the number, size, quality of seeds, number of capsules, and number of undeveloped flowers, respectively. Standardized patch area, patch proximity, and their interactions were included as explanatory variables. To account for potential cross pollination from wild bluebells, the occurrence of wild bluebells was also included as an additional covariate in all the models (either 0 or 1; categorical variables and recorded in 10 of the 51 woodlands). Model residuals were plotted and visually checked using 'DHARMa' package [55] in R [v 4.0.2, 56]. Mann-Whiteney $U$ tests were used to determine the effect of open pollination on the number of seeds, number of seed capsules, and number of flowers that failed to set seeds, comparing differences between the pollinator exclusion plants kept in the greenhouse and the plants placed in the woodland patches.

## Results

### Species-area relationship for the overall plant community

There were a total of 106 plant species recorded in the 51 woodland patches at the time of sampling, with 41 species recorded as flowering (S1 Table in S1 File). The number of plant species in each habitat varied from 3 to 41 species. As expected, given the predictions of island biogeography theory, the overall number of species found in the patch significantly increased with patch area ($z = 0.34$, $p < 0.001$, Fig 2A) and when considered separately, the number of spring flowering plants also positively related with patch area ($z = 0.20$, $p = 0.002$, Fig 2B),

### The effect of patch area and proximity on pollination

Compared with the bagged plants, open pollination increased the number of bluebell seeds by 741% (Mann-Whitney $U$ test, W = 88, $p = 0.001$) and the number of capsules by 331% (Mann-Whitney $U$ test, W = 111, $p = 0.003$). There was no effect of open pollination on the number of flowers that did not set seed though (Mann-Whitney $U$ test, W = 361, $p = 0.762$, Fig 3).

We found that both the number of seeds and seed capsules are positively related to the patch proximity. (Table 1 and Fig 4). However, no relationship was found between patch proximity and the size and quality of seeds (Table 1). Patch area and presence of wild bluebells had no effects on seed set and seed capsules (Table 1). Finally, there was no evidence of an interaction between patch area and proximity on the number, size, quality of seeds, number of seed capsules, and undeveloped flowers (Table 1).

## Discussion

Understanding the impact of patch area and isolation is an important endeavour if we are to conserve pollinators and pollination against habitat loss and fragmentation. Although previous studies have demonstrated the impact of habitat loss and fragmentation in a variety of systems

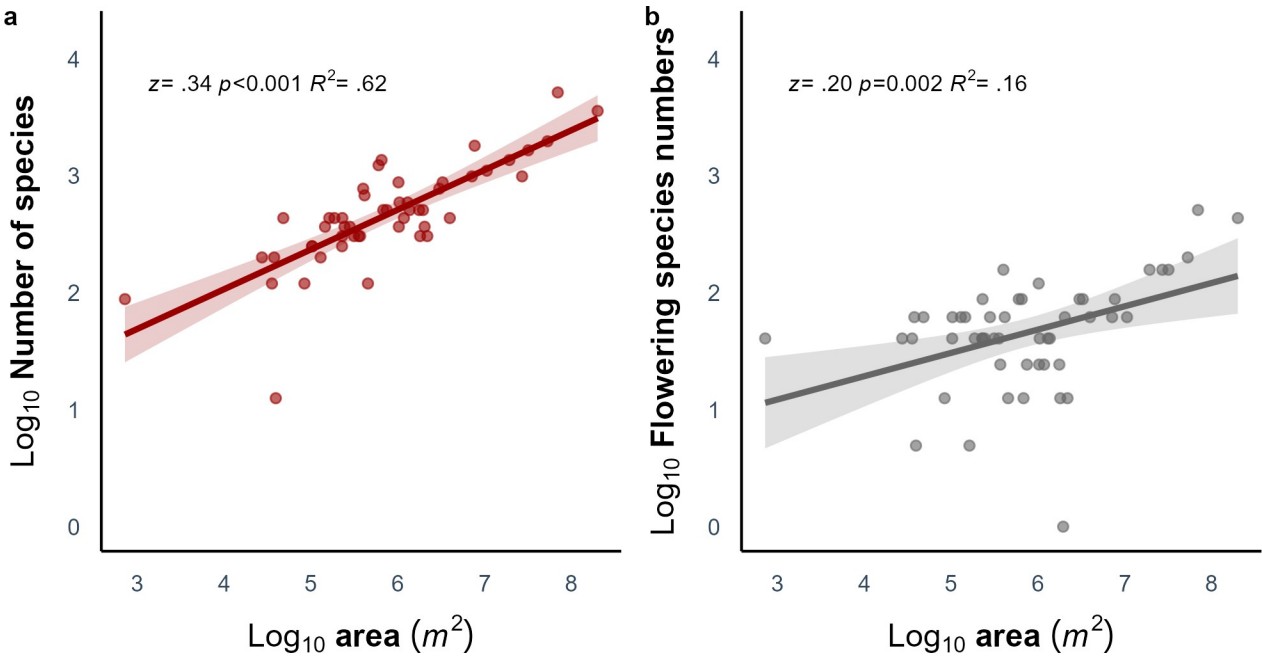

**Fig 2.** The species-area relationship of a) total number of plant species and b) number of spring flowering species. A linear model was shown in a solid line, with 95% confidence intervals (CIs). Numbers of species (points) were $\log_{10}$ transformed.

[e.g., 13, 24], we still have a rather limited understanding on the effect of fragmentation, and how it interacts with habitat loss, to affect spring pollination. In our study landscape, the backdrop to our experiment is a significant positive relationship between total number of plant species and habitat area, indicating that our woodland patches are a genuinely fragmentated system [19]. By placing phytometer bluebell plants in our fragmented landscape, we found that whilst patch area had little effect on bluebell pollination, isolation significantly reduced its reproductive success, as both seed number and seed capsules decreased with increasing distance between patches.

There are three limitations to our study. First, our experimental scale is relatively small (study site <1.7 km$^2$), compared with previous studies examining the effect of fragmentation in much larger landscapes [e.g., 30]. However, this means that our results are a relatively conservative estimate of the impact of fragmentation, thus the effect could be greater on a larger scale, as the movements of pollinators are potentially more affected. Our understanding of pollination services could also be improved if this experiment were replicated over years. Second, while bluebells are known to be predominantly visited by bumblebee queens which other taxa visited our experimental plants remains unknown [but see DoPI database for recorded bluebell visitors in the UK, 57]. Although these phytometer bluebells were unlikely visited by plenty of non-bee pollinators during this time of the year, it is possible for a small number of early insects such as pollen beetles (e.g., *Meligethes.*) to visit the plant. If bluebells were also visited by other insects, fragmentation may impact pollination by differentially affecting the movements of different pollinators. For instance, isolation may be more detrimental to the pollination of less mobile pollinators than bumblebees [58]. Ideally, testing this requires a community level approach [e.g. 59] where plant communities that attract different types of pollinators are used as a bioassay of pollination efficacy. In addition, measuring actual pollinator activity along with the community of phytometer plants would improve our understanding of how plant-pollinator networks respond to habitat change. Third, it is important to note that other

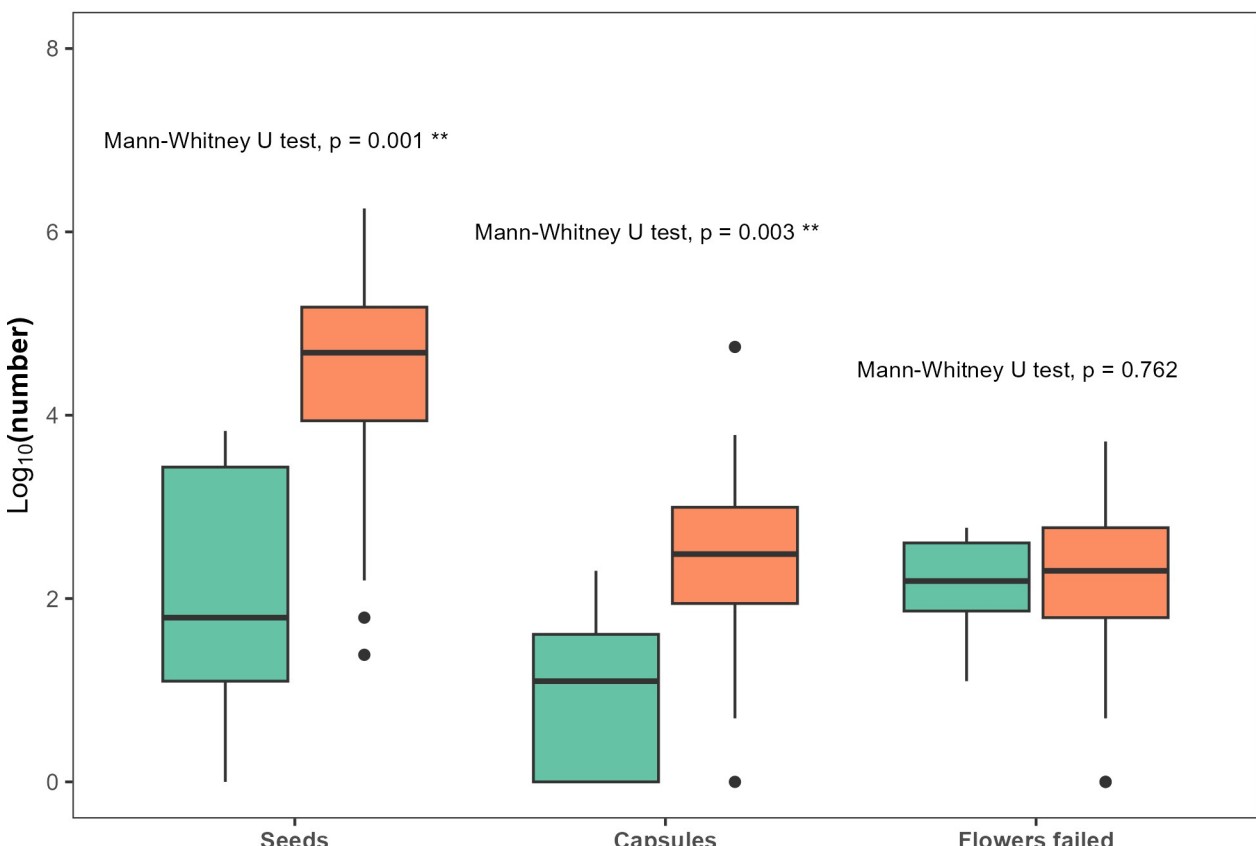

**Fig 3. The effect of bagged flowers (sample size n = 6 pots, with three bluebell plants in each) versus open-pollination (sample size = 130 pots, i.e. the plants placed in the woodland patches) on the number of seeds and seed capsules and the number of flowers that failed to set any seeds.** Numbers were $\log_{10}$ transformed. Significant levels: 0 '***', 0.001 '**', 0.01 '*', 0.05.

abiotic factors, such as variation in microclimatic variables [15, 60], may affect plant-pollinator dynamics especially in small fragmented system. This may be particularly important to pollinators in early spring, as some pollinators may have limited thermoregulation abilities which makes them unable to operate at the lower temperatures inside woodland fragments. Incorporating microclimatic conditions with habitat configurations may be important to consider in this pollination system.

**Table 1. Summary of fitted GLMs on the effect of patch area, proximity, wild bluebells, and the interaction of patch area and proximity on the number, size, quality of seeds, and number of seed capsules and undeveloped flowers.** Significant $p$ values were highlighted in bold. Significant levels: 0 '***', 0.001 '**', 0.01 '*', 0.05.

| | no. of seeds | | size of seeds | | quality of seeds | | no. of seed capsules | | no. undeveloped flowers | |
|---|---|---|---|---|---|---|---|---|---|---|
| | estimate | p-value | estimate | p-value | estimate | p-value | estimate | p-value | estimate | p-value |
| Intercept | 54.415 | **0.046** * | 1.854e+00 | **<0.001** *** | 55.240 | **<0.001** *** | 5.098 | 0.107 | 11.732 | **<0.001** *** |
| Area | 1247.152 | 0.092 | -1.541e+00 | 0.618 | 2.960 | 0.690 | 163.447 | 0.061 | 23.541 | 0.645 |
| Proximity | 0.241 | **0.005** ** | -1.856e-05 | 0.957 | 0.0003 | 0.809 | 0.032 | **0.002** ** | 0.001 | 0.893 |
| Wild bluebells | -33.489 | 0.220 | 2.911e-02 | 0.800 | 0.044 | 0.156 | -4.319 | 0.177 | 1.896 | 0.320 |
| Area: proximity | -5.367 | 0.118 | 9.753e-03 | 0.498 | -0.017 | 0.424 | -0.632 | 0.117 | -0.178 | 0.454 |

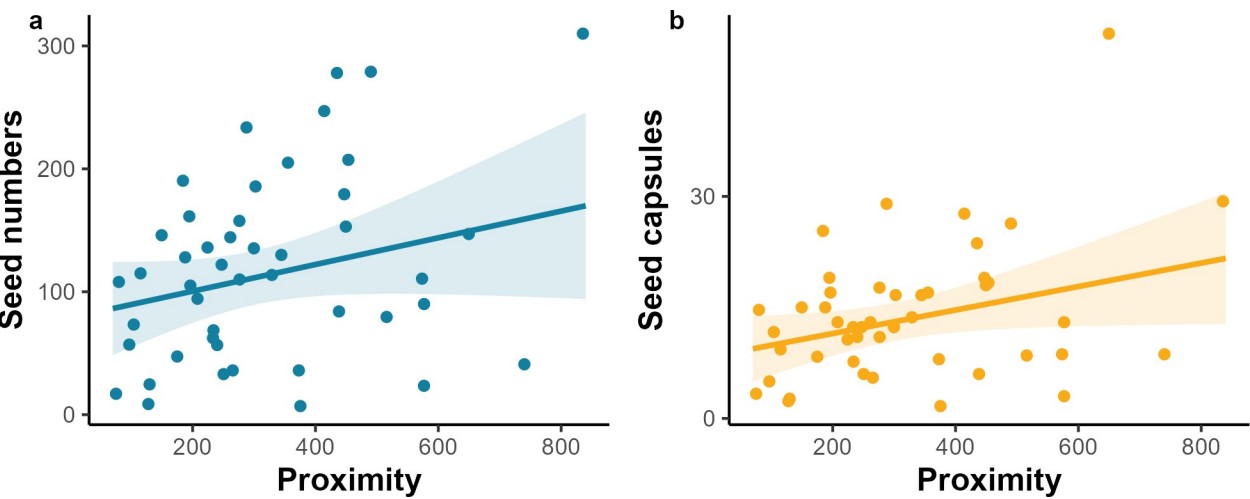

**Fig 4.** The effect of patch proximity (i.e., the opposite of isolation) on the number of seeds (a) and seed capsules (b) of bluebells. Dots represent data points and lines are model outputs, with 95% CIs. Significant levels: 0 '***', 0.001 '**', 0.01 '*', 0.05.

Our results showed that both seed set and capsule development of bluebells increased with proximity, a phenomenon which largely agrees with previous findings that isolation is the key factor limiting pollination services in our system [e.g. 61–63]. We interpret these results by the fact that patch isolation reduced pollinator visitation and thus pollen exchange, causing lower reproductive outcomes of this species. A previous study found that bluebells were largely self-incompatible and produced more seeds and capsules with cross pollination [53], and we also showed here that open pollination produced more seeds per capsule than pollinator exclusion. Thus, it is likely that more isolated patches had reduced pollinator activity and had a lower probability of cross pollination, similar to the findings in other studies [63, 64]. Notably, although the purpose of this study was to disentangle how patch area and isolation differently affected pollination, it is possible that closer patches with a high proximity may function as a large habitat for pollinators. This means that pollinators could freely move if there were more available patches nearby. Overall though, our results suggest that habitat fragmentation negatively impact the pollination of bluebells, by increasing isolation between habitat patches.

In addition to the number of seeds produced by plants, pollination may affect seed formation by influencing their size and/or quality [65]. This has been demonstrated in other systems [e.g., 66]. Surprisingly, the size and quality of seeds were unaffected by any of the factors we measured, suggesting that either the seed maturation of bluebells is either not limited by the quality of pollen received or that, providing they are visited, isolated plants receive sufficient pollen from a low visitation rate. That said, we did not measure germination rate, which would be a better measure of seed quality. A recent study showed that *H. non-scripta* strongly prevented autogamous and geitonogamous pollination by rejecting pollen tube growth to the end of the ovule styles, resulting in lower seed set in contrast to cross pollination [52]. Collectively, these results may indicate that female reproduction of bluebells is more likely to be affected by the quantity of pollen grains from cross pollination, rather than the quality of cross pollination. To this end, reducing habitat isolation and thereby allowing more pollen dispersal should be a priority when conserving bluebells and similar types of flowering plants.

Large habitat patches generally host more pollinators than small patches by inherently providing more diverse resources for pollinators to forage [67]. In our study, we measured the contribution of pollinator activities to seed set, by placing same number of phytometer

bluebells in woodland patches irrespective of their sizes, meaning that plants are likely to set more seeds in large patches if they are limited by pollen supply [12, 68]. However, we demonstrated that this may not be applicable to spring flowering plants, as both seeds and seed capsules of bluebells were unaffected by patch area in our study. We attribute this to insufficient pollinators in spring, which potentially results in a pollen limitation of bluebells in regardless of patch size. Previous studies found that spring pollinators in forest fragments were strongly associated with the availability of understory foraging resources [40], and some habitats, like forest edges, may temporally host more pollinators in spring due to better light exposure [41, 42]. This may lead to fewer pollinator visitations for woodland specialists such as bluebell. In our study, plants in small patches may receive similar visitation with large patches, and this may be true when some insect pollinators are able to forage particular plant species over the whole landscape [69]. In fact, some dominant spring pollinators, like bumblebees, were able to optimize foraging rewards by visiting fewer flowers in large patches [70]. Our results suggest that large habitat patches may not necessarily lead to more pollination, especially for plants that highly rely on the availability of spring pollinators.

In summary, our study demonstrates that for bluebells at least, habitat isolation is a greater threat to pollination than habitat size. It also highlights the importance of small habitat patches and connectivity in maintaining pollination services. As spring pollination is particularly susceptible to climatic disturbances [44], finding an effective strategy to conserve populations of spring flowering plants is important. Reducing isolation, for example by using pollinator corridors, may improve the fitness of spring flowering plants and increase their population resilience. From a practical perspective, these results shed some light on the mechanisms underlying the effect of habitat fragmentation on pollination services and provide some pointers for landscape managers as to the best approaches for conserving the pollination of early spring flowers.

## Supporting information

**S1 File. Supporting information of isolation limits spring pollination in a UK fragmented landscape.**
(DOCX)

## Acknowledgments

We thank Bristol City Council who allowed access to the site. We thank Nina Bosch Fernandez, Lily Adeniji, and Ellie Nichols for providing field and lab assistance.

## Author Contributions

**Conceptualization:** Dongbo Li, Jane Memmott.

**Data curation:** Dongbo Li.

**Formal analysis:** Dongbo Li.

**Investigation:** Dongbo Li.

**Supervision:** Christopher F. Clements, Jane Memmott.

**Writing – original draft:** Dongbo Li.

**Writing – review & editing:** Christopher F. Clements, Jane Memmott.

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
