## [Decision Letter · Decision Letter 0]

15 May 2024

PONE-D-24-05394Isolation limits spring pollination in a UK fragmented landscapePLOS ONE

Dear Dr. Li,

Thank you for submitting your manuscript to PLOS ONE. After careful consideration, we feel that it has merit but does not fully meet PLOS ONE’s publication criteria as it currently stands. Therefore, we invite you to submit a revised version of the manuscript that addresses the points raised during the review process.

We look forward to receiving your revised manuscript.

Kind regards,

Daniel de Paiva Silva, Ph.D.

Academic Editor

PLOS ONE

Journal Requirements:

D. L was funded by the China Scholarship Council (grant no. 20186190011)

6. We note that Figure 1 and S1 in your submission contain map and satellite images which may be copyrighted. All PLOS content is published under the Creative Commons Attribution License (CC BY 4.0), which means that the manuscript, images, and Supporting Information files will be freely available online, and any third party is permitted to access, download, copy, distribute, and use these materials in any way, even commercially, with proper attribution. For these reasons, we cannot publish previously copyrighted maps or satellite images created using proprietary data, such as Google software (Google Maps, Street View, and Earth). For more information, see our copyright guidelines: http://journals.plos.org/plosone/s/licenses-and-copyright.

We require you to either present written permission from the copyright holder to publish these figures specifically under the CC BY 4.0 license, or remove the figures from your submission:

a. You may seek permission from the original copyright holder of Figure 1 and S1 to publish the content specifically under the CC BY 4.0 license.  

Additional Editor Comments:

Dear Dr. Li,

After this first review round, we obtained two positive reviews and one negative one. I believe that if you are able to solve the issues raised by all reviewers, your manuscript will be suitable for publication in PLoS One.

Sincerely,

Daniel Silva

Reviewers' comments:

Reviewer's Responses to Questions

**Comments to the Author**

1. Is the manuscript technically sound, and do the data support the conclusions?

Reviewer #1: Yes

Reviewer #2: Partly

Reviewer #3: Yes

2. Has the statistical analysis been performed appropriately and rigorously? 

Reviewer #1: Yes

Reviewer #2: No

Reviewer #3: Yes

3. Have the authors made all data underlying the findings in their manuscript fully available?

Reviewer #1: Yes

Reviewer #2: Yes

Reviewer #3: Yes

4. Is the manuscript presented in an intelligible fashion and written in standard English?

Reviewer #1: Yes

Reviewer #2: Yes

Reviewer #3: Yes

5. Review Comments to the Author

**Reviewer #1**: In this study, the authors investigated the effect of habitat size and isolation on the plant-pollinator interaction. To do so, they conducted pollination experiments using potted plant phytometry, in which they placed English Bluebell. The work and experimental design attempt to separate the effect of habitat size and isolation on the plant-pollinator interaction. The authors conclude that the data suggest isolation of areas poses a greater threat to pollination than habitat size. It's a very elegant study with a great idea and well-written.

However, I have a few minor queries.

Although not explicitly stated, it appears that the authors placed the same number of English bluebell individuals in all patches, regardless of area size and isolation. First, I would like to request the authors to clarify whether the number of bluebell individuals was the same for all areas or not, in a clear manner. Second, if the number of plants was the same for all areas regardless of size and isolation, I have some doubts.

The authors correctly point out that larger areas should have a greater abundance of pollinators and that this could increase reproductive success in these areas. However, the data found show that the reproductive success of bluebells was not impacted by area size. The authors attribute this to insufficient pollinators, which could result in pollen limitation for the plants. I agree that the observed result is likely due to pollen limitation, but I have a different perspective on this. I wonder if areas of different sizes receiving the same number of bluebell individuals could affect these results? I say this because pollination is a process dependent on the density of pollinators (as the authors state in the text), but also on the density of conspecific plants. Larger areas should have a greater number of plant individuals, which would increase the chances of pollen transfer. I wonder if the experimental design masked this relationship. When I see the result that reproductive success was associated with proximity, I wonder if the closer areas are close enough to function like a larger area with more individuals?

Results

L 218-219 - How many individuals were flowering?

**Reviewer #2:** The authors evaluated the effect of the proximity and size of habitat patches on the reproductive success of a plant species largely pollinated by early spring bumblebee queens, as a phytometer plant.

The study has some limitations, which were carefully elucidated by the authors in the discussion, but which still represent an alternative for evaluating the effects of habitat fragmentation on pollination services in subsequent studies, considering a less local scale.

Below are some considerations about this study.

Line 49: of flowering plants rely on animal pollination

Line 85: In most cases

Line 101: foraging and nesting

Line 130: Do these flowering plant species share bee visitors with English bluebells?

Line 134: Could you improve the resolution quality of figure 1?

Line 168-170: During the experiment, 9 plants were placed in 51 patches, totaling 459 plants. Of these, only 130 plants were successful in survival and pollination, representing less than 30% of the number of plants used at the beginning of sampling.

Was there a total loss of plants in any of the sampling patches? If so, were these areas included or removed from subsequent analyses?

Line 212: The Kruskal-Wallis test is a non-parametric test used to compare the variation of a numerical variable with a categorical variable of at least 3 groups. According to the data presented by the authors, the numeric variables (i.e. the number of seeds, number of seed capsules, and number of flowers that failed to set seeds) were compared using two categories of plants: plants exposed to pollinators vs plants prevented from pollinators.

If this is the case, the analysis could be carried out using a more appropriate non-parametric test considering the nature of the data in the present study.

Line 226-230: See previous comment on the section "Data analysis"

Line 234: We found that both the number of seeds and seed capsules are positively related to the patch proximity. (Table 1, Fig. 4). However, no relationship was found between patch proximity and the size and quality of seeds

**Reviewer #3: **Overall, the authors have written a very clear and concise manuscript that provides interesting information on how habitat patch isolation impacts pollination within a relatively small study area in the spring. This information will help future studies that are looking to determine what factors are limiting pollination within various landscapes, including small fragmented habitats. Overall I don't have many comments and all my comments below are minor revisions.

Introduction

- It would be helpful to discuss the other factors outside of habitat fragmentation and isolation that might impact the different plant pollination metrics being measured, for example microclimatic variables such as temperature and light.

- Lines 107-108: Add references to what spring fragmentation studies have been completed to date and refer back to their results within the discussion.

Methods and materials

- It would be good to elaborate on how the plant surveys were conducted within the patches, for example how many transects/quadrats were used per area? Was number of flowering plants also measured or just species richness?

Collecting Seed Data

- Lines 187-188: Not clear how sub-samples were chosen, it would be good to include the number of subsamples analyzed per plant

Data Analysis

- The number of plant species was found to be correlated with area, but assessing whether the number of flowering plants in total was correlated with area would also be helpful. Was this data measured within the plant surveys? There may have been some patches will small amounts of area, but a relatively high number of flowering plants utilized by spring pollinators (for example flowering trees/shrubs) which may explain why area wasn’t significantly associated with pollination. If plant abundances are available, I would suggest investigating their correlations with area and possible the pollination metrics (e.g., seed set).

Results

- Line 221 – the p-value should be put within the brackets for this sentence

Discussion

- No line numbers

- A reference should be provided to prove that a significant positive relationship between total number of plant species and habitat area indicates that the woodland patches are a genuinely fragmentated system.

- It would be good to acknowledge that future studies could be improved by measuring pollinator visitation in addition to the metrics measured in this study so information on the relationships between pollinators and pollination could be further expanded on.

- The discussion should speak further to what other factors could reduce seed set and capsule development outside of reduced pollination, for example microclimatic variables, for example light and temperature

6. PLOS authors have the option to publish the peer review history of their article (what does this mean?). If published, this will include your full peer review and any attached files.

Reviewer #1: No

Reviewer #2: No

Reviewer #3: No

---

## [Author Response · Author response to Decision Letter 0]

25 Jul 2024

(Response letter also has been attached in a separate file). 

Dear Dr. Daniel de Paiva Silva,

We have resubmitted a revised copy of our manuscript entitled “Isolation limits spring pollination in a UK fragmented landscape” (PONE-D-24-05394).

We greatly appreciate the time and effort that editors and reviewers have dedicated to providing insightful comments. We have incorporated all the comments provided by reviewers, and thank you for the additional time we were given to do this. Specifically, we first have updated our manuscript to meet journal’s style requirements, and corrected issues found in data analysis. Finally, we have provided extra information in discussion to address comments raised by reviewers. 

In addition, we have proof-read the whole manuscript to improve the overall quality of publication. The major changes we have made have been highlighted in yellow.

Many thanks for considering our manuscript and we hope it now acceptable for publication in PlOS ONE.

Sincerely,

Dongbo Li (On behalf of all the authors)

RESPONSE TO EDITOR

Authors’ response: We have re-formatted the paper to make it consistent with journal style requirements, including the title page, heading font sizes in the main text, figure names and resolutions. 

Authors’ response: We have added information in lines 134-135. 

Authors’ response: We have made all the software codes and data free to access in figshare data repository(https://doi.org/10.6084/m9.figshare.26046196) and GitHub repository (https://github.com/Dongboli/experimental-data). We have included this in the data availability statement in the MS in lines 26-28. 

D. L was funded by the China Scholarship Council (grant no. 20186190011)

Authors’ response: We have included this in the funding statement in lines 22-24. 

Authors’ response: We have updated data availability statement on page 2 (lines 26-28), allowing all the software codes and data to be freely available for peer-review and publication. 

6. We note that Figure 1 and S1 in your submission contain map and satellite images which may be copyrighted. All PLOS content is published under the Creative Commons Attribution License (CC BY 4.0), which means that the manuscript, images, and Supporting Information files will be freely available online, and any third party is permitted to access, download, copy, distribute, and use these materials in any way, even commercially, with proper attribution. For these reasons, we cannot publish previously copyrighted maps or satellite images created using proprietary data, such as Google software (Google Maps, Street View, and Earth). For more information, see our copyright guidelines: http://journals.plos.org/plosone/s/licenses-and-copyright.

 Authors’ response: We thank the editor for providing some useful resources. We have replaced the original S1 Figure that contains a satellite map with an aerial photography provided under CC BY 4.0 license. The basemap in Fig 1 was recreated using ‘GADM’ open-source data with specific permission for publication in PLOS (see https://gadm.org/license.html). We have updated the figure legend accordingly in lines 149-150. 

Authors’ response: We have checked the reference list to make sure they were correctly cited. There were no retracted papers in the reference list. 

RESPONSE TO REVIEWER 1: 

1) Although not explicitly stated, it appears that the authors placed the same number of English bluebell individuals in all patches, regardless of area size and isolation. First, I would like to request the authors to clarify whether the number of bluebell individuals was the same for all areas or not, in a clear manner.

Authors’ response: The number of bluebell individuals was the same for all the habitat patches, regardless of patch size and isolation. We have made in clear on the manuscript in line 189. 

2) If the number of plants was the same for all areas regardless of size and isolation, I have some doubts. The authors correctly point out that larger areas should have a greater abundance of pollinators and that this could increase reproductive success in these areas. However, the data found show that the reproductive success of bluebells was not impacted by area size. The authors attribute this to insufficient pollinators, which could result in pollen limitation for the plants. I agree that the observed result is likely due to pollen limitation, but I have a different perspective on this. I wonder if areas of different sizes receiving the same number of bluebell individuals could affect these results? I say this because pollination is a process dependent on the density of pollinators (as the authors state in the text), but also on the density of conspecific plants. Larger areas should have a greater number of plant individuals, which would increase the chances of pollen transfer. I wonder if the experimental design masked this relationship. When I see the result that reproductive success was associated with proximity, I wonder if the closer areas are close enough to function like a larger area with more individuals?

Authors’ response: We agree with the reviewer that plant density is important to the chances of pollen transfer, which may affect reproductive success of plants in the patches. However, as we are more interested in how the size and isolation of woodland patches affect pollinator activities and therefore pollination, same densities of bluebell plants were used as a controlled phytometer, in which cases that phytometer plants should produce more seed set if there were more pollinator visitations, under controlled plant densities. Indeed, if greater numbers of plants were placed within larger habitat patches, we could not distinguish whether the higher seed set was resulted from patches with higher pollinator activities, or patches with higher flower densities. We have included this in the discussion in lines 339-341.

We agree with the reviewer that a cluster of patches could potentially function as a large patch if they are close enough, meaning that higher seed set could be a result of more phytometer plants occurred in that habitat. However, as we were aiming to disentangle how patch area and isolation differently affected pollination, woodland patches with different sizes and positions were selected as experimental arena (Fig 1a). We calculated proximity index based on the accumulative degree of isolation of one patch to the rest of all the patches (i.e., the rest 50 patches in the total 51 patches), weighted by their sizes. In that case, we could separate the effect of patch area and isolation by treating each woodland patch as an individual observation, through placing the same density of phytometer plants. Although it was not very clear if a cluster of closer patches could function as a larger patch having more pollen flow, our results suggested that the proximity to other patches was more important than the actual size of patch. We also have discussed this associated with proximity index in lines 319-323. 

L 218-219 - How many individuals were flowering?

Authors’ response: There were totally 106 plants recorded, with 41 species recorded as flowering. We have included this in lines 243-244. 

RESPONSE TO REVIEWER 2:

1) Line 49: of flowering plants rely on animal pollination

Authors’ response: This has been corrected in lines 56-57. 

2) Line 85: In most cases

Authors’ response: This has been corrected in line 95. 

3) Line 101: foraging and nesting

Authors’ response: This has been updated in line 110.

4) Line 130: Do these flowering plant species share bee visitors with English bluebells?

Authors’ response: We have added this information in lines 145-146. 

5) Line 134: Could you improve the resolution quality of figure 1?

Authors’ response: Figure 1 has been updated in a higher resolution. 

6) Line 168-170: During the experiment, 9 plants were placed in 51 patches, totaling 459 plants. Of these, only 130 plants were successful in survival and pollination, representing less than 30% of the number of plants used at the beginning of sampling.

Was there a total loss of plants in any of the sampling patches? If so, were these areas included or removed from subsequent analyses?

Authors’ response: We originally placed 153 pots of 459 plants in 51 patches, and collected 130 pots of 390 plants (i.e. 130 x 3 = 390) at the end of experiment. Two patches that had completely no plants collected (i.e., three pots of 9 plants were all lost) were excluded from further analysis. We had made it clear in lines 194-196.

7) Line 212: The Kruskal-Wallis test is a non-parametric test used to compare the variation of a numerical variable with a categorical variable of at least 3 groups. According to the data presented by the authors, the numeric variables (i.e. the number of seeds, number of seed capsules, and number of flowers that failed to set seeds) were compared using two categories of plants: plants exposed to pollinators vs plants prevented from pollinators.

If this is the case, the analysis could be carried out using a more appropriate non-parametric test considering the nature of the data in the present study.

Authors’ response: We agree with the reviewer that using Kruskal-Wallis test is inappropriate to the data, thus we’ve re-analysed our data using Mann-Whitney U test and updated in lines 236-237, 254-256 and Fig 3. In brief this did not result in any changes to overall conclusion. 

8) Line 226-230: See previous comment on the section "Data analysis"

Authors’ response: The results have been updated in lines 254-256.

9) Line 234: We found that both the number of seeds and seed capsules are positively related to the patch proximity. (Table 1, Fig. 4). However, no relationship was found between patch proximity and the size and quality of seeds

Authors’ response: The statements have been corrected in lines 261-263. 

RESPONSE TO REVIEWER 3: 

1) Introduction

- It would be helpful to discuss the other factors outside of habitat fragmentation and isolation that might impact the different plant pollination metrics being measured, for example microclimatic variables such as temperature and light.

Authors’ response: We have included this in the introduction in lines 70-74.

2) Lines 107-108: Add references to what spring fragmentation studies have been completed to date and refer back to their results within the discussion.

Authors’ response: We have added this in the introduction in lines 114-120, and discussed our results with other studies in the discussion in lines 346-350. 

3) Methods and materials

- It would be good to elaborate on how the plant surveys were conducted within the patches, for example how many transects/quadrats were used per area? Was number of flowering plants also measured or just species richness?

Authors’ response: We conducted field survey by taking random samples of all ground vegetation to record the overall richness of plant species in each patch, and flowering status of each species. We have included this in lines 172-176. 

4) Collecting Seed Data

- Lines 187-188: Not clear how sub-samples were chosen, it would be good to include the number of subsamples analyzed per plant

Authors’ response: We measured seed sizes by taking a small portion of total seeds produced each pot and took photographs (where one subsample would allow seeds to be photographed in a high image quality). We have included this in lines 207-212. 

5) Data Analysis

- The number of plant species was found to be correlated with area, but assessing whether the number of flowering plants in total was correlated with area would also be helpful. Was this data measured within the plant surveys? There may have been some patches will small amounts of area, but a relatively high number of flowering plants utilized by spring pollinators (for example flowering trees/shrubs) which may explain why area wasn’t significantly associated with pollination. If plant abundances are available, I would suggest investigating their correlations with area and possible the pollination metrics (e.g., seed set).

Authors’ response: We agree that we should account for variation in spring flowering plants among patches, which may be correlated with habitat area. We have investigated the relationship between spring flowering plants with area in lines 218, 247-248, and standardized patch area with number of flowering species in analysing pollination metrics in lines 227-230, and updated our results and Table 1 (page 15). In brief we found that the number of spring flowers was positively related with patch area (lines 247-248), but accounting for the number of spring flowering plants with patch area did not change the results that patch area had no effect on pollination metrics measured in our study (lines 261-263, Table 1). 

6) Results

- Line 221 – the p-value should be put within the brackets for this sentence

Authors’ response: We have added p values in lines 247-248. 

7) Discussion

- No line numbers

Authors’ response: Line numbers have been added in discussion on pages 16-29. 

8) A reference should be provided to prove that a significant positive relationship between total number of plant species and habitat area indicates that the woodland patches are a genuinely fragmentated system.

Authors’ response: We have added a reference in lines 282-283.

9) It would be good to acknowledge that future s

---

## [Decision Letter · Decision Letter 1]

5 Sep 2024

Isolation limits spring pollination in a UK fragmented landscape

PONE-D-24-05394R1

Dear Dr. Li,

We’re pleased to inform you that your manuscript has been judged scientifically suitable for publication and will be formally accepted for publication once it meets all outstanding technical requirements.

Kind regards,

Daniel de Paiva Silva, Ph.D.

Academic Editor

PLOS ONE

Additional Editor Comments (optional):

Dear Dr. Li,

I am pleased to accept you manuscript for publication in PLoS One!

Sincerely,

Daniel Silva

Reviewers' comments:

Reviewer's Responses to Questions

**Comments to the Author**

1. If the authors have adequately addressed your comments raised in a previous round of review and you feel that this manuscript is now acceptable for publication, you may indicate that here to bypass the “Comments to the Author” section, enter your conflict of interest statement in the “Confidential to Editor” section, and submit your "Accept" recommendation.

Reviewer #1: All comments have been addressed

Reviewer #3: All comments have been addressed

2. Is the manuscript technically sound, and do the data support the conclusions?

Reviewer #1: Yes

Reviewer #3: Yes

3. Has the statistical analysis been performed appropriately and rigorously? 

Reviewer #1: Yes

Reviewer #3: Yes

4. Have the authors made all data underlying the findings in their manuscript fully available?

Reviewer #1: Yes

Reviewer #3: Yes

5. Is the manuscript presented in an intelligible fashion and written in standard English?

Reviewer #1: Yes

Reviewer #3: Yes

6. Review Comments to the Author

Reviewer #1: (No Response)

Reviewer #3: The authors have addressed all my previous comments adequately and I have no further comments. Overall, this study provides interesting information on how habitat patch isolation impacts pollination within a relatively small study area in the spring. This information will help future studies that are looking to determine what factors are limiting pollination within different landscapes and provides helpful guidance on how future studies can improve upon the methods used within this study.

7. PLOS authors have the option to publish the peer review history of their article (what does this mean?). If published, this will include your full peer review and any attached files.

Reviewer #1: No

Reviewer #3: No

---

## [Editor Report · Acceptance letter]

9 Sep 2024

PONE-D-24-05394R1 

PLOS ONE

Dear Dr. Li, 

I'm pleased to inform you that your manuscript has been deemed suitable for publication in PLOS ONE. Congratulations! Your manuscript is now being handed over to our production team.

Kind regards, 

on behalf of

Dr. Daniel de Paiva Silva 

Academic Editor

PLOS ONE